# The Information Conveyed in a SPAC′s Offering

**DOI:** 10.3390/e23091215

**Published:** 2021-09-15

**Authors:** Gil Cohen, Mahmoud Qadan

**Affiliations:** 1Department of Management, Western Galilee Academic College, Acre 2412101, Israel; 2School of Business Administration, University of Haifa, Haifa 3498838, Israel; Mqadan@univ.haifa.ac.il

**Keywords:** SPACS, IPO, information, pipes, market efficiency

## Abstract

The popularity of SPACs (Special Purpose Acquisition Companies) has grown dramatically in recent years as a substitute for the traditional IPO (Initial Public Offer). We modeled the average annual return for SPAC investors and found that this financial tool produced an annual return of 17.3%. We then constructed an information model that examined a SPAC′s excess returns during the 60 days after a potential merger or acquisition had been announced. We found that the announcement had a major impact on the SPAC’s share price over the 60 days, delivering on average 0.69% daily excess returns over the IPO portfolio and 31.6% cumulative excess returns for the entire period. Relative to IPOs, the cumulative excess returns of SPACs rose dramatically in the next few days after the potential merger or acquisition announcement until the 26th day. They then declined but rose again until the 48th day after the announcement. Finally, the SPAC’s structure reduced the investors’ risk. Thus, if investors buy a SPAC stock immediately after a potential merger or acquisition has been announced and hold it for 48 days, they can reap substantial short-term returns.

## 1. Introduction

The SPAC (Special Purpose Acquisition Company) is a relatively new financing tool initiated at the beginning of 2016. Since then, it has grown rapidly in terms of the number of SPAC Initial Public Offers (IPOs) and the amount of money they raise. The U.S. SEC (Securities and Exchange Commission) defines SPACs as a development stage company that has no specific business plan or purpose or has indicated that its business plan is to engage in a merger or acquisition with an unidentified company or companies. SPACs have become increasingly popular vehicles for entities or individuals to raise capital for merger opportunities, and for private companies seeking to raise capital, obtain liquidity for existing shareholders and become publicly traded. SPACs are also called blind capital pools or “Private Investment Public Equity” (PIPES). While there is no maximum size of a target company, there is a minimum size (roughly 80% of the funds in the SPAC’s trust account). Hence, a relatively small company would not be a suitable acquisition candidate for a relatively large SPAC unless combined with another sizable target. Occasionally, SPACs acquire multiple small targets simultaneously, where the individual targets are not sufficiently large to justify a bilateral transaction. SPACs typically seek to combine with target companies that have a value of two to four times the amount of their IPO proceeds to reduce the diluting impact of the founders’ shares and warrants.

In this research, we constructed a structural SPAC model and found that it produces an average annual return of 17.3%. Moreover, we challenged the “Random Walk” and “Efficient Market” hypothesis (Discussed later) by testing the information conveyed to the financial markets in the SPACs’ announcement of a potential business combination. If the financial market is efficient, such an announcement should be absorbed immediately and not continue to influence the stock′s price afterwards. We chose a 60-day window after a merger opportunity announcement to examine how fast the new information is absorbed into the SPAC’s share price. Our benchmark for measuring excess returns and risk is a traditional IPO portfolio. The risk involved in investing in a SPAC versus an IPO is similar in nature because they both invest in firms that are taking their first steps in the financial markets. However, a SPAC’s announcement of a potential business combination differs from traditional IPO fundraising procedures in the sense that the SPAC investors must confirm the suggested future merger. As a result, it involves more risk to investors who must weigh the chances of such a business combination being approved.

When a SPAC is initiated, the owners usually state that they seek to find a suitable business combination within a specific sector. That is the first information conveyed to the financial markets by the SPAC initiators i.e., identifying potential sectors and activities that may create value to investors. The second time that information was conveyed by the SPAC to the financial markets is the announcement that a potential merger partner has been found and an agreement has been signed. At this important stage, the market applaud the fact that the SPAC’s goals are going to be fulfilled. The last conveyed information to the financial markets is the merger’s actual approval by the SPACs shareholders. In this paper, we are concentrating on the second stage of information transferred by the SPAC since we see this stage as the most important stage in the evolution of the SPAC from a capital company to a firm that has real assets and business model.

We found that, on average, these 60 days produce 0.69% daily excess returns over the IPO portfolio and 31.6% cumulative excess returns for the entire period. We also demonstrated that the volatility of the SPAC’s price after a merger announcement is greater than that of the IPO portfolio. However, this extra volatility can be diversified away using many such assets.

## 2. Literature Review

SPACs are companies that go public with the sole purpose of acquiring or merging with a private firm by using the money that was previously raised through an initial public offering (IPO). SPACs are listed shell corporations with the purpose of acquiring a private company, thus making them public without going through the traditional initial public offering process. Although SPACs are taking an essential share of the IPO market, the phenomenon has not yet been fully explored. Previous studies have focused on the unique structure and features of SPACs Lakicevic et al. [1], Chatterjee et al. [2], Boyer and Baigent [3], Okutan Nilsson [4], Cumming et al [5], and Vulanovic [6]. Other studies have investigated the performance of SPACs in various market conditions (for example: Kolb and Tykvova [7], and Dimitrova [8]). Blomkvist and Vulanovik [9] examined the wave pattern of U.S. SPACs using data from 2003. In their principal analysis, they used two dependent variables. First, they defined the SPACs’ share as the proportion of quarterly SPAC IPOs scaled by the total of IPOs. Second, they used the number of quarterly SPACs to determine the volume of the SPACs. They found that both the volume and share of the SPACs relative to the total of IPOs were negatively related to a market measure of uncertainty (the VIX) and time-varying risk aversion. They concluded that the SPAC’s sponsor can credibly signal the issue’s quality through the purchase of additional warrants.

A SPAC’s merger with another firm changes the capital structure of the target firm. Kopecky et al. [10] demonstrated that the existence of such entities gives rise to a viable alternative equilibrating process that we dub “equity-price shifting.” This alternative no-arbitrage process is straightforward and leads to the same optimal firm value. In that sense, in our view, the future change in the optimal capital market should be priced into the SPAC’s share price immediately after the announcement of a potential merger. Combining the SPAC with the target firm creates a new firm whose management is comprised of individuals from both firms. Coles and Lee [11] examined the importance of managers’ heterogeneity in determining the primary aspects of the executives’ incentives, the firm’s policy, its risk and its performance. The identity of the CEO of the target company may provide important information to the SPAC’s shareholders because the CEO’s preferences and past performance are made public when the company is about to be publicly traded.

Song and Shin [12] argued that investors regard more opaque investments as risker. This opacity also increases their level of risk aversion. The information channel postulates that investors’ willingness to participate in SPACs offering is negatively related to time-varying uncertainty and risk aversion, creating demand fluctuations. Using data from all 87 SPAC IPOs from 2003 until 2006, Boyer and Baigent [3] demonstrated that the SPAC’s structure enables investors to buy units that are risk-free securities due to the creation of an escrow account and their ability to withdraw the entire investment. Some researchers have been skeptical about SPACs and concluded that they will underperform the market, on average, in the long run. Kolb and Tykvová [7] analyzed 127 SPAC acquisitions and 1128 IPOs during the wave of “new-generation” SPACs starting in 2003. Their findings supported the conjecture that small and leveraged firms with limited growth opportunities tend to use this vehicle. In their view, SPAC acquisitions may also be fueled by the cash-out motives of existing shareholders. Moreover, venture capitalists and private equity investors tend to refrain from using SPAC acquisitions as an exit route. Another interesting finding is that SPAC firms are associated with severe underperformance compared to the market, the industry and comparable IPO firms.

To explore more about the performance of SPACs, we focused on a 60-day window after the SPAC announced that a principal agreement had been signed between the SPAC and the target company. In our view, this specific period is extremely important in terms of the new information conveyed to the financial markets concerning the possible realization of the SPAC′s goal to merge with a company that contains real business assets and a business growth strategy. We also used our findings to test the “Random Walk” hypothesis, which is a consequence of the Efficient Market Hypothesis. It posits that stock prices reflect all relevant and available information, and that it is not possible to consistently outperform the market using this information. The finance literature attempts to explain patterns in stock market returns as evidence of the behavior of irrational investors. Empirically, various market anomalies such as the January effect have been observed in stock markets worldwide (see, for example, Jones [13], Kampman [14]) challenged the “Random Walk” hypothesis. Zhang and Zheng [15] explained the randomness of stock price movements as a reaction to mixed market signals or in terms of unanticipated changes in investors’ preferences.

## 3. Analysis and Data

SPACs have a set of regulations that they must follow. First, they have a maximum of three years to find a business combination opportunity. If they do not find an appropriate business partner in that timeframe, the investors get their money back with interest. Second, the SPAC′s investors have the right to vote on the upcoming acquisition. In addition, at the time of merger, the SPAC is required to offer shareholders the option to redeem their stock for $10 (The $10 value is the common regulated amount). These terms create a structural financial asset with an intrinsic value of $10 and two possible outcomes for the limited period of three years. The first outcome (1-P) could be the return of the investors’ capital, meaning that they have lost only the value of their time and the eroded value of their investment due to inflation. The second outcome (P) may be a business combination announcement that occurs before the limited time has expired (t) that drives the SPAC’s share price to $Z upon announcement of a possible merger. Figure 1 illustrates these possible occurrences.

The financial possibilities described in Figure 1 are presented in Equation (1). Solving for *R* will produce the mean annual return.
(1)$$10=P\times \frac{Z}{{\left(1+R\right)}^{t}}+\left(1-P\right)\times \frac{10}{{\left(1+R\right)}^{3}}$$
where Z is the price of the share after the merger has been approved, and *t* is the time to the merger’s approval.

Figure 2 shows the distribution of SPACs and non-SPAC IPOs. Figure 3 illustrates the different development stages of SPACs from 2009 until April 2011.

From Figure 2, we learn that SPAC IPOs accounted for 33% of all IPOs in 2020. In contrast, SPACs were only 14.6% of all IPOs in 2016. These numbers indicate the growing popularity of SPACs as a means of providing cash for IPOs.

Figure 3 indicates that from 2009 until April 2021, 1041 SPACs were established. During that period, 196 mergers were completed, 126 SPACs announced that they found a suitable merger opportunity and only 26 SPACs were liquidated. Excluding SPACs that still had time to find a merger opportunity (within the 3-year limit), 9.5 of all of the SPACS were liquidated (https://www.spacanalytics.com/). Moreover, according to SPAC Insider (https://spacinsider.com/stats/) the average time for a SPAC to find an appropriate merger is a year and a half. The average return from a SPAC IPO after a merger has been announced is 27%, including the warrants’ valuation. Integrating this data into Equation (1), we get Equation (2).
(2)$$10=\left(0.905\right)\frac{12.7}{{\left(1+R\right)}^{1.5}}+\left(1-0.905\right)\frac{10}{{\left(1+R\right)}^{3}}$$
solving for *R* results in an average annual return of 17.3%.

We used data about the daily returns of 20 SPACs that had recently announced that they had found a suitable merger for a future business combination. It is important to note that we did not include in our sample any SPACs that announced that they were exploring certain possible combinations. All of our SPACs announced that they had found a target firm and signed conditional agreements to be approved by both companies’ shareholders. Furthermore, to be included in our sample, all of our SPACs had to have made only one announcement of this sort since their creation. Such criteria allowed us to exclude market speculation about potential mergers that eventually crumbled and identify only real information conveyed by a signed agreement that was officially announced by the SPAC.

We followed the stock prices’ reactions for 60 days after the announcement day. In addition, we collected comparable data for an IPO Exchange Trade Fund (ETF) so we could compare the returns of the traditional fundraising methodology (IPO) and the SPACs.

## 4. Conditional Entropy and the Chain Rule

In information theory, conditional entropy quantifies the amount of information needed to describe the outcome of a random variable given that the value of another random variable is known. The entropy of *Y* conditioned on *X* is written as H(Y ⋮ X). The chain rule assumes that the combined system determined by two random variables *X* and *Y* has joint entropy H(X,Y). If we first learn the value of *X*, we have gained H(X) information. Once X is known, we only need H(X,Y)−H(X) to describe the state of the whole system. This quantity is exactly H(Y ⋮ X), which constitutes the chain rule of conditional entropy (Cover and Thomas [16]).
(3)H(Y ⋮ X)=H(X,Y)−H(X)

The launching of a SPAC contains information that in a specific sector should be a good investment opportunity. For example, a SPAC that declares upon its initiation that it is going to seek a merger opportunity in the fintech field indicates that the board of directors of this SPAC feels that this field of operation provides future investment opportunities. The second signal generated by a SPAC occurs when it declares that it has found a suitable merger opportunity in its area of expertise. In doing so, the SPAC indicates that it will probably achieve the desired merger in the allowed time frame and will not be liquidated. Moreover, the announcement of the upcoming merger delivers information from the SPAC leaders to investors that they believe that the merger will increase the value of the combined company.

We tested whether the SPAC’s merger announcement could serve as a chain rule of conditional entropy for future price changes 60 days after the announcement. The proposition behind abnormal returns upon announcing an upcoming merger challenges the famous “Random Walk” hypothesis that argues that historic price changes are unable to predict future price changes as given in Equation (4).
(4)$$\left(HU_{t1\ldots t60}\;\vdots \;DU_{t}\right)=HU_{t1\ldots t60}$$
where $HU_{t1\ldots t60}\;\vdots \;DU_{t}$ is the probability of an up-price trend on days t1…t60 given that the SPAC announces a potential merger opportunity on day t.

The “Random Walk” hypothesis is consistent with the “Efficient Market” hypothesis in their contentions that future price changes are impossible to predict using past information. This theory suggests that changes in stock prices have the same distribution and are independent of each other. Therefore, it is impossible to outperform the market without assuming additional risk. However, if a merger announcement can predict future price trends in the near period afterwards, we would expect that the probability of a price uptrend would rise after a merger announcement and that the signaling effect would fade over time (Equations (5) and (6)).
(5)$$\left(HU_{t1\ldots t60}\;\vdots \;DU_{t}\right)>HU_{t1\ldots t60}$$
(6)$$\left(HU_{t1}\;\vdots \;DU_{t}\right)>\left(HU_{t2}\;\vdots \;DU_{t}\right)>\cdots \left(HU_{t60}\;\vdots \;DU_{t}\right)$$
where $HU_{t1\ldots t60}\;\vdots \;DU_{t}$ is the probability of an up-price trend on days t1…t60 given the SPAC’s announcement of a potential merger opportunity on day t.

We tested for abnormal returns by comparing individual SPAC returns to an IPO ETF, which represents a broad range of IPOs companies. If price abnormality exists after the SPAC’s business combination announcement and if it fades over time, we expect that the cumulative abnormal returns of SPACs relative to those of the IPO ETF would rise in the days after the announcement and decline over time.

## 5. Findings

Figure 4 and Figure 5 depict the cumulative daily returns of two of our sample SPACs (tickers: SHLL and GRAF) versus the IPO ETF for 60 days after the announcement of an upcoming business combination with a target company.

The figures indicate substantial differences in the cumulative daily returns of the SPAC stocks versus the IPO ETF. In both cases the SPACs performed much better than the IPO portfolio after the announcement of the future merger. SHLL achieved 16.28% higher cumulative returns over the 60 days and GRAF gained 85.62% more cumulative returns over that period. We define excess returns (ER) and cumulative excess returns (AER) for 60 consecutive days after a potential business combination has been announced (Equations (7) and (8)).
(7)$$ER=SPAC_{K}-IPO_{K}$$
(8)$$AER=\overset{60}{\underset{k=1}{\sum }}\mathrm{SPA}C_{K}-IPO_{K}$$
where: $\mathrm{SPA}C_{K}$ = the SPAC’s returns on day *k* and $IPO_{K}$ = the IPO’s returns on day *k*. The potential business combination was announced on day *k* = 0.

Table 1 summarizes the returns, excess returns and cumulative excess returns of the 20 sampled SPACs over 60 days after the merger announcement versus the traditional fundraising mechanism (IPO) for the equivalent period.

Table 1 indicates that the average time that elapses from the launching of a SPAC in our sample and the announcement of a potential business combination was 506 days. The shortest period was 75 days (FVAC) and the longest was 1092 days (LACQ). On average, the SPACs in our sample earned a daily return of 0.99% during the 60-day period, whereas the IPO’s average was only 0.30%. Thus, the SPACs earned 0.69% percent excess returns over the IPO portfolio. The highest daily excess returns were for SHLL (2.75%) followed by DPHC (2.27%). On average, the standard deviation for the 60-day period of the SPACs was 6.52 for the SPACs and only 2.16 for the IPO. Thus, during the 60 days after a potential merger announcement, the SPAC stocks exhibited greater volatility than the IPO ETF, caused by the investors attempting to evaluate the merits of the proposed business combination. Calculating using the Sharp ratio, we get 0.152 for the SPACs and 0.138 for the IPO, suggesting a better risk-return ratio for the SPACs than for the IPO.

During the 60-day period, the SPACs stocks had a cumulative return of 49.09% versus only 17.48% for the IPO portfolio, resulting in 31.6% in cumulative excess returns, as illustrated in Figure 6.

As the figure indicates, once a merger has been announced, the SPAC’s price rises dramatically, achieving on average more than 10% cumulative excess returns a few days after the announcement day and increasing to 31.6% cumulative excess returns for the 60-day period. The figure also shows that a setback in the cumulative excess returns occurs from day 26 until day 40 and from day 48 until day 60. The results summarized in Table 1 and Figure 6 support Equations (5) and (6), meaning that conditional entropy does exist when a SPAC announces a potential merger. The announcement delivers information to current and potential investors that the opportunity to gain real value is about to occur. The future business combination between the SPAC, which is the funding company, and the target company, which has actual business assets, should appreciate to a point that exceeds the sum of values of the individual companies. Moreover, the effect of the new information that is conveyed in the merger announcement is very strong in the days following the announcement, but fades over time. This phenomenon signals that the new information is slowly absorbed into the financial market and reaches its full extent 48 days after the announcement.

Finally, we utilized a linear regression that used the IPO portfolio as the market index in the framework of the famous CAPM (Capital Assets Pricing Model) (See for example Wu and Chiou [17]) theory to examine excess returns and risks. Equation (9) formulates the theoretical model and Equation (10) the corresponding results.
(9)$$SPAC_{K}-RF_{K}=\alpha +\beta \left(IPO_{k}-RF_{K}\right)$$
(10)$$SPAC_{K}-RF_{K}=0.848+0.762\left(IPO_{k}-RF_{K}\right)$$
T Stat; (3.71)   (7.59).
$$R^{2}=0.06,\;Fsig=0.001$$
where: $SPAC_{K}-RF_{K}$ is the SPAC’s return on day *k* minus the corresponding daily risk-free return. $IPO_{k}-RF_{K}$ is the IPO’s return on day *k* minus the corresponding daily risk-free return. α measures the excess returns of the SPACs over the IPO’s returns and β is the systemic risk factor.

Equation (10) provides evidence that the CAPM framework also supports the contention that SPACs provide positive and significant excess returns compared to the IPO portfolio. Moreover, the systemic risk factor represented by β is less than 1, indicating less systemic risk for SPACs versus IPOs. Although, as Table 1 illustrates the total risk of investing is SPACs during the 60 days is greater than investing in the IPO ETF, the regression line indicates that there is less systemic risk.

## 6. Summary and Implications

The popularity of SPACs has grown dramatically in recent years as a substitute for the traditional IPO. The strict regulations that govern it should reduce the risk to investors. We started our analysis by characterizing these restrictions and regulations. We then modeled the average annual return for SPAC investors, demonstrating that it produced 17.3%. Next, we constructed an information model that measured the information embedded in the SPAC′s announcement of an upcoming business combination with a potential firm. The expected merger would involve the properties of both original companies, increasing the potential value of the resulting firm. Moreover, the business combination would change the target company’s capital structure and might also alter other aspects such as risk preferences and future business models. These prospective changes are absorbed into the SPAC’s share price after the announcement of an upcoming business combination, making the information that such an announcement carries critical information for the future of the joint company. We demonstrated that that information had a major impact on the SPACs’ shares prices for 60 days after it was delivered to the financial markets. On average, these SPACs had 0.69% daily excess returns over the IPO portfolio and 31.6% cumulative excess returns for the entire period. Moreover, even though the SPACs’ returns during the 60 days were more volatile than those of the IPO returns, they had less systemic risk than the IPO. Therefore, investors can reduce their investment risks using diversification. Furthermore, the cumulative excess returns of the SPACs over the IPO rose dramatically in the next few days after the announcement of the potential merger until the 26th day. Then, there was a drop but a renewal of the increase until the 48th day following the merger’s announcement.

We conclude that the SPAC’s structure reduces investors’ risks. This investment vehicle can also produce substantial short-term returns if investors buy the SPAC’s stock immediately after a potential merger has been announced and hold it for 48 days. These results imply that SPAC investors and regulatory authorities should better regulate announcements about potential business combinations to differentiate between real future business combinations and trial balloons designed to test investors’ behavior. These results should also be retested in the future using a large number of SPACs and different regulations that are constantly changing as this important, relatively new, financial asset evolves.

## Figures and Tables

**Figure 1 entropy-23-01215-f001:**
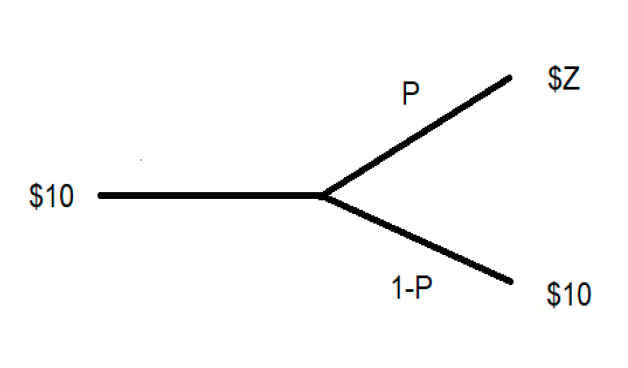
Possible outcomes of the SPAC’s price.

**Figure 2 entropy-23-01215-f002:**
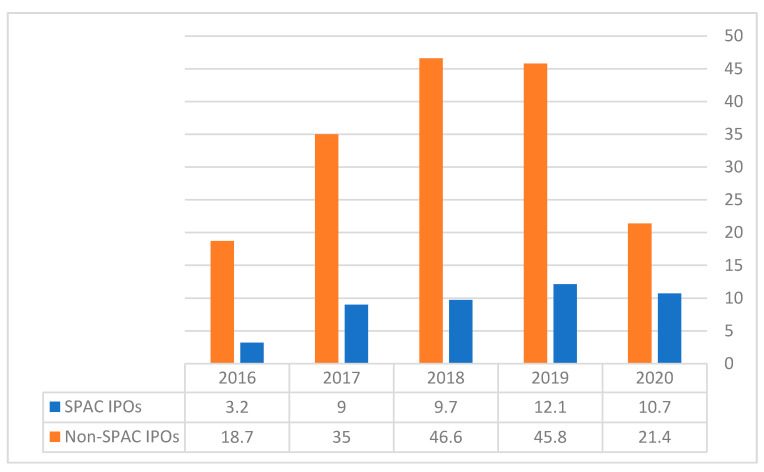
SPACs and non-SPAC IPOs in billions of dollars. Source: VitalSigns IPO Data.

**Figure 3 entropy-23-01215-f003:**
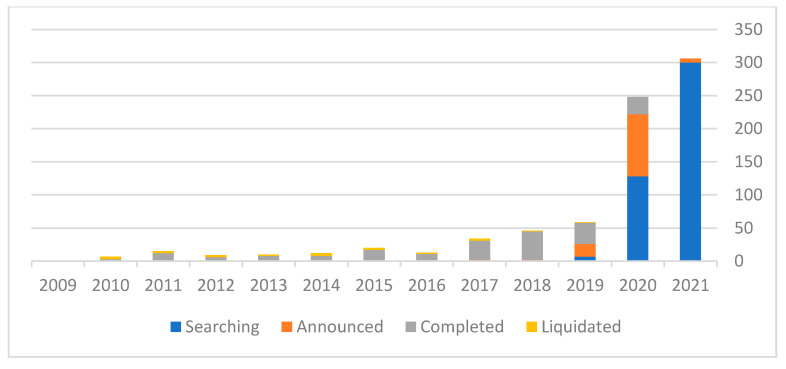
Different development stages of SPACs from 2009 until April 2011.

**Figure 4 entropy-23-01215-f004:**
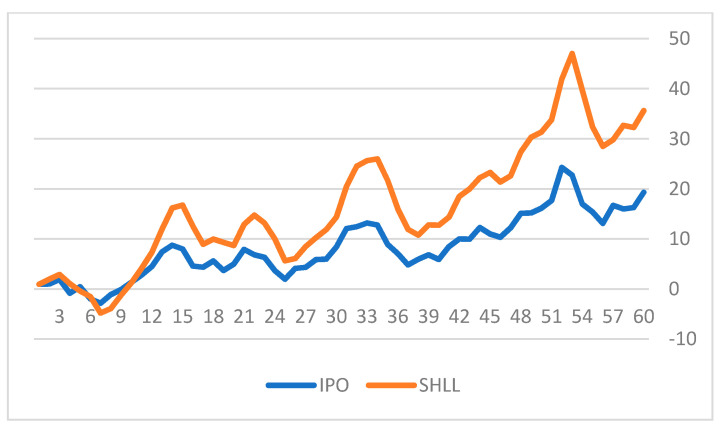
Cumulative Returns of SHLL and IPO.

**Figure 5 entropy-23-01215-f005:**
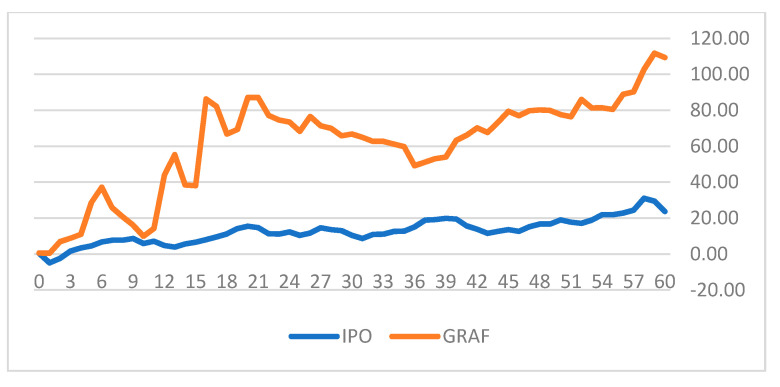
Cumulative returns of GRAF and IPO.

**Figure 6 entropy-23-01215-f006:**
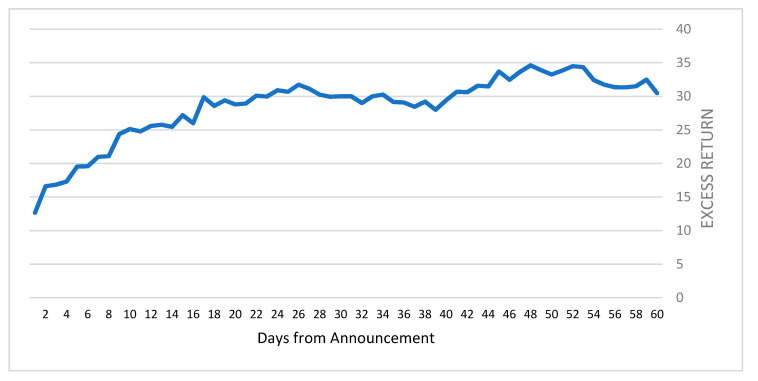
Cumulative excess returns of SPACs over IPO portfolio.

**Table 1 entropy-23-01215-t001:** 60-Day SPAC and IPO returns after a merger announcement.

	SPAC’s Ticker	Days Before Merger	SPAC’s Daily Average Returns (1)	IPO’s Daily Average Returns (2)	SPAC’s Cumulative Returns (3)	IPO’s Cumulative Returns (4)	(1–2)	(3–4)
1	DPHC	519	2.67% (7.18)	0.40% (2.27)	109.64%	16.8%	2.27%	92.84%
2	SHLL	475	3.07% (11.36)	0.32% (2.12)	35.61%	19.33%	2.75%	16.28%
3	FVAC	75	0.75% (4.48)	0.38% (2.17)	40.81%	22%	0.37%	18.81%
4	GRAF	603	1.70% (9.76)	0.53% (2.06)	109.35%	23.73%	1.17%	85.62%
5	FMCI	662	0.85% (6.25)	0.41% (1.97)	50.96%	24.54%	0.44%	26.42%
6	DMYT	126	0.50% (2.33)	0.45% (2.07)	30.22%	26.87%	0.05%	3.35%
7	LCA	416	1.08% (7.38)	0.43% (2.08)	65.12%	26.03%	0.65%	39.09%
8	INSU	446	0.45% (3.94)	0.30% (2.11)	27.6%	22.57%	0.15%	5.03%
9	SPAQ	698	0.71% (7.13)	0.30% (2.14)	42.16%	18.60%	0.41%	23.56%
10	DKNG	167	0.32% (5.93)	−0.42% (3.71)	19.27%	−25.23%	0.74%	44.5%
11	OPES	815	0.25% (5.23)	0.43% (1.91)	16.10%	22.20%	−0.18%	-6.1%
12	CFFA	551	0.0% (1.14)	0.32% (2.11)	0.05%	19.59%	−0.32%	-19.54%
13	LACQ	1092	0.95% (13.12)	0.10% (2.01)	57.32%	6.32%	0.85%	51%
14	TDAC	906	0.52% (3.63)	0.49% (1.89)	31.40%	28.32%	0.03%	3.08%
15	ALAC	798	0.06% (4.83)	−0.12% (2.49)	4.39%	−1.35%	0.18%	5.74%
16	THCB	625	1.40% (10.17)	0.42% (1.90)	82.80%	26.26%	0.98%	56.54%
17	APXT	429	0.70% (5.06)	0.42% (1.90)	40.58%	26.26%	0.28%	14.32%
18	SRAC	381	1.51% (7.06)	0.30% (2.01)	82.51%	17.93%	1.21%	64.58%
19	WPF	173	0.14% (2.44)	0.52% (1.85)	9.15%	30.56%	−0.38%	-21.41%
20	CCIV	162	2.08% (11.97)	−0.06% (2.43)	126.78%	−1.57%	2.14%	128.3%
Average (St. Dev)		506	0.99% (6.52)	0.30% (2.16)	49.09%	17.48%	0.69%	31.60%
Sharp Ratio			0.152	0.138				
Max		1092	3.07%	0.53%	126.78%	30.56%	2.75%	128.3%
Min		75	0.0%	−0.42	0.05%	−25.23%	−0.18%	−21.41%

Notes: SPAC’s and IPO’s Daily Average Return = the average daily return of the examined 60 days period. SPAC’s and IPO’s Cumulative Returns = the 60 days cumulative return of SPACs and IPOs.

## Data Availability

Upon Request.

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
