# Peer review of "The Information Conveyed in a SPAC′s Offering"

_entropy, 2021, doi:10.3390/e23091215_

Round 1
Reviewer 1 Report
The authors study the evolution of SPACs share price over the sixty days period after the announcement of a potential merger. The subject is interesting, but I find some issues. The main ones are the following:
- After introducing section 4, one might expect to apply it in section 5 (Findings), but it does not. Are the hypothesis in section 4 (equations (5) and (6)) satisfied for the SPACs covered? An analysis of conditional entropy for the covered SPACs will be interesting, otherwise section 4 seems more like an excuse to write something related to entropy.
- There are only 20 SPACs covered, so I don't think the findings are conclusive for all SPACs, so the conclusions in the summary section should be taken with caution. On the other hand, it is not entirely clear which criteria were used to select those 20 SPACs and no other. Are they just the most recent, as suggested, or are there other criteria?
- A very plausible explanation for the abnormal returns of the SPACs is that, in the period analyzed, the SPACs have more volatility than the stocks included in the IPO ETF, due to the uncertainties of the announcement. Some measure, such as the Sharpe ratio, should be used to check whether or not this is the case.
- In lines 213-218, some conclusions are obtained from equation (10), but it is reported that R2 = 0.06 for the linear regression, so the conclusions do not seem significant to me.
Some minor issues:
- Conditional entropy should be defined in the paper, so it is more self-contained.
- There is two sections 5.
Author Response
The reply is attached

Reviewer 2 Report
In this paper the authors analyse the if Special Purpose Acquisition Companies (SPAC) have any difference in the effect on excess return, when compared with regular Initial Public Offers (IPO). The results show that the excess return is higher in the case of SPAC. The motivation of the paper is interesting, although I have several issues that I would like to suggest authors to rethink and reflect, in particular:
- In the abstract, acronyms appear without being identified firstly. It happens with SPAC and IPO. And in the introduction with SEC.
- Authors refer some times to "on average 0.69 daily excess return over the IPO". My question is: 0.69 what? It is missing the unit, right?
- The literature review is scarce and it misses, clearly, a better description of the methodology which are used in those studies. It should be described those methodologies, in order to understand their differences.
- Moreover, I believe that sections 1 and 2 are too short but maybe the authors should manage them jointly, identifying also clearly the way how your proposal is better.
- In line 55 authors identify market-wide uncertainty as VIX. Note that VIX is not the definition of market uncertainty but one possible way to measure it.
- The way as SPAC regulations are identified with 1. (line 69) and 2. (line 71) are not organized properly and according to the guidelines.
- The option of the 10$ described in line 73 is always the same? Or is used in your model only?
- Authors should homogenize the way they present results. For example, in line 91 it appears "percent" and in the following line "%"
- Authors talk about random walk and efficient market hypotheses without any reference. These are mainstream theories which request some discussion. It is important to discuss the way has the work of the authors is important for that hypotheses. And authors also need to manage some works about those hypotheses in the literature review.
- Why does authors use the 60 days as a benchmark? Why not 100, for example?
- Equation 9 comes from where? Some reference? And how is it estimated? Through OLS? Issues like stationarity and structural breaks were taken in account? An none is said about the r-square?
- It is also missing larger discussion about the results and its real implications for the different market agents.
- The paper has several gramatical errors, identifiable also by me, which I do not have a perfect English. Please, make or request a final proofreading.
Author Response
reply attached

Reviewer 3 Report
attached

Author Response
reply is attached

Round 2
Reviewer 1 Report
I do not feel that the authors have answered my main issues:
- After introducing section 4, one might expect to apply it in section 5 (Findings), but it does not. Are the hypothesis in section 4 (equations (5) and (6)) satisfied for the SPACs covered? An analysis of conditional entropy for the covered SPACs will be interesting, otherwise section 4 seems more like an excuse to write something related to entropy.
I definitively do not understand the role of section 4. The analysis of conditional entropy for the covered SPACs will justified it, but it has not been done.
- There are only 20 SPACs covered, so I don't think the findings are conclusive for all SPACs, so the conclusions in the summary section should be taken with caution. On the other hand.
The sample is too small to obtain general conclusions. However, you can obtain partial conclusions for the covered SPACs or the period analyzed.
- A very plausible explanation for the abnormal returns of the SPACs is that, in the period analyzed, the SPACs have more volatility than the stocks included in the IPO ETF, due to the uncertainties of the announcement. Some measure, such as the Sharpe ratio, should be used to check whether or not this is the case.
The result for the Sharpe ratio (0.152 vs 0.138) does not seem too significant. They are quite similar, so this explains to some extend the abnormal returns (you are just taking a greater risk).
- In lines 213-218, some conclusions are obtained from equation (10), but it is reported that R2 = 0.06 for the linear regression, so the conclusions do not seem significant to me.
If R2=0.06, the linear regression is not significant, so the values of alfa and beta are not relevant.
Some minor issues:
- Conditional entropy should be defined in the paper, so it is more self-contained.
The definition of H(X) and H(X,Y) is still missing.
Reviewer 2 Report
I believe that the authors didn't make enough to clarify all the issues. For example, considering my original comments:
3. The literature review is scarce and it misses, clearly, a better description of the methodology which are used in those studies. It should be described those methodologies, in order to understand their differences.
9. Authors talk about random walk and efficient market hypotheses without any reference. These are mainstream theories which request some discussion. It is important to discuss the way has the work of the authors is important for that hypotheses. And authors also need to manage some works about those hypotheses in the literature review.
10. Why does authors use the 60 days as a benchmark? Why not 100, for example?
11. Equation 9 comes from where? Some reference? And how is it estimated? Through OLS? Issues like stationarity and structural breaks were taken in account? An none is said about the r-square?
All of them are not correctly discussed/explained. For example, it is not saying that more 5 papers were added that solve the problem. Or that the Random Walk and Efficient Market Hypothesis were addressed that the issue is solved. How is it possible to identify something about EMH without citing the original work, for example). The same happens with the answer of my concern number 10: saying that "We used 60 days as our "window" time frame since it is commonly used time frame in financial event studies". Which studies?
And finally, maybe more important, I know that equation 11 is the one of CAPM. The problem is not that, but the possible problems with the estimation, which are nor identified neither discussed.
Reviewer 3 Report
Well done.